# MorphOcc: An implicit generative model of neuronal morphologies

## Abstract

Understanding the diversity and complexity of the morphology of different types of neurons is important for understanding neural circuits. We need quantitative, unbiased methods to capture the structural and morphological features of neurons. With the advent of large-scale structural datasets, this analysis becomes feasible using data-drive approaches. Existing generative models are limited to modeling dendritic and axonal skeleton graphs, without considering the actual 3D shape. In this work, we propose MorphOcc, a model that represents the diversity of neurons in mouse primary visual cortex (V1) in a single neural network by encoding each neuron's morphology into a low-dimensional embedding. From this embedding the 3d shape can be reconstructed. We train our model on 797 dendritic shapes of V1 neurons. The learned embedding captures morphological features well and enables cell type classification into known cell types. Interpolating between samples in embedding space generates new instances of neurons without supervision. MorphOcc has the potential to improve our understanding of neurons in the brain by facilitating large-scale analysis and providing a model for representing neuronal morphologies.

## 1 Introduction

The diversity of neuronal morphologies has fascinated researchers for over 100 years (Ramón y Cajal, 1911). Understanding a neuron's structure is important, because it constraints the functions the neuron can implement. For example, the length and branching patterns of dendrites and axons affects the way that neurons receive and transmit signals (Goldberg et al., 2004; Hill et al., 2012; Oberlaender et al., 2012). Neurons vary significantly in their morphology: Some neurons have long, massively branching dendrites, which allow them to receive and integrate information from many other cells. Other neurons have a more compact structure, with dendrites and axons that are shorter and less branched (Markram et al., 2015; DeFelipe et al., 2013).

Starting with early work by Cajal Ramón y Cajal (1911), we have learned a great deal about the different morphological cell types in the brain and some of the core principles of their morphological organization. However, much of this knowledge is based on visual inspection (Ramón y Cajal, 1911; Defelipe et al., 2013) or manually defined features such as the length and branching patterns of their dendrites and axons (Scorcioni et al., 2008; Armañanzas & Ascoli, 2015; Wang, 2018; Kanari et al., 2019; Gouwens et al., 2019), but does not describe the full heterogeneity and morphological diversity of cell types. To capture this full diversity, we would need generative models that can sample realistic instances of neurons. Previous work on generative models exist, for instance using biologically motivated growth rules (van Pelt & Schierwagen, 2004; Eberhard et al., 2006), manipulating shape templates until they approximately match the observed data (Cuntz et al., 2011; Farhoodi & Kording, 2018) or 3D random walks (Laturnus & Berens, 2021). However, all of these methods generate tree-like representations – skeletons of neurons – and therefore do not generate details beyond the skeletonal graph.

We propose MorphOcc, an implicit model for neuronal morphologies that allows clustering and neuron generation at the same time. Our model captures the diversity of cells in a single network and embeds the 3D shapes into low dimensional latent vectors. These latent codes – or "bar codes" – are used to cluster cell types and retrieve neurons. We further use the latent codes to reconstruct the neurons. By interpolating between two neurons' latent codes, we generate new morphologies

that resemble the previously seen neurons. The analysis on 797 neurons shows that our clustering is consistent with existing knowledge on cell types and MORPHOCC has the potential to reveal new findings in neuroscience.

## 2 RELATED WORK

3D objects are represented in various ways. We differentiate between explicit and implicit methods. The most common explicit representations are meshes, point clouds and voxels. Meshes describe 3D objects with faces and vertices. Point clouds are a natural choice for representing 3D data acquired from scanning sensors such as LiDAR or depth cameras. Voxels represent 3D objects in a grid-like structure of values. Additionally, neurons tend to be skeletonized into graph-like structures to reduce data complexity.Those skeletons consist of nodes and edges with features, e.g. Cartesian coordinates as node features.

There exist several approaches on generating skeletonized neuronal morphologies. One approach is growing tree-like structures based on biologically motivated growth rules (van Pelt & Schierwagen, 2004; Memelli et al., 2013; Torben-Nielsen & De Schutter, 2014; Koene et al., 2009; Palombo et al., 2019). Ascoli et al. (2001) and Eberhard et al. (2006) delevoped software tools (L-Neuron and Neu-Gen) to generate morphologies based on recursive and descriptive, iterative rules, respectively, to model the growth of dendritic patterns of neurons. Such methods are limited to generating neurons according to known rules, but by definition cannot discover new cell types or principles of morphological organization from data. Kanari et al. (2022) introduce a topology-guided synthesis algorithm that generates neurons where they sample topology values and then use a dendritic growth algorithm. Other approaches manipulate shapes until they approximately match the observed data, i.e. by first sampling points or morphologies followed by iterative perturbation (Cuntz et al., 2011; Farhoodi & Kording, 2018). MorphVAE (Laturnus & Berens, 2021) generates neural morphologies using a sequence-to-sequence variational autoencoder that operates on 3D-walks within the tree structure of a neuron and then heuristically combines the random walks into a complete neuron morphology. However, all those methods generate tree-like representations – skeletons – of neurons which do not contain any detail of the neuron, such as the thickness or local curvature of the dendrites.

In contrast to relying on explicit data structures to representing signals, implicit methods represent signals by parameterizing a mapping $f(\mathbf{x})$ where $\mathbf{x}$ is a location in space (and potentially time) and $f(\mathbf{x})$ are signal properties at $\mathbf{x}$. In the context of 3D shapes, $\mathbf{x}$ is a point in $\mathcal{R}^3$ and $f(\mathbf{x})$ indicates the location of a point relative to the surface of the object. This results in continuous, memory-efficient representations of the 3D geometry of neurons without topological restrictions. Traditionally, implicit approaches address representing only a single object or scene (Sitzmann et al., 2020b; Takikawa et al., 2021; Martel et al., 2021; Müller et al., 2022). While early approaches used simple MLPs Mescheder et al. (2019); Chen & Zhang (2018), more recent work incorporated mechanisms to increase detail. This involves Fourier features and periodic activation functions Sitzmann et al. (2020b). Other approaches are based on a distributed feature volume, which can be generated by encoding an image Saito et al. (2019); Xu et al. (2019), be distributed across an octtree Takikawa et al. (2021) or an implicit grid Jiang et al. (2020). These approaches do not fit our purpose as they only represent a single sample. Furthermore, we require the presence of a representative vector (shape code) for each object to enable clustering. Multi-shape representation has been shown by occupancy networks (Mescheder et al., 2019) and IM-Net (Chen & Zhang, 2018). However, the methods only work on relatively simple shapes, such as objects in ShapeNet (Chang et al., 2015) and have not been shown to represent fine details of the objects. DeepSDF (Park et al., 2019) introduce a latent code-conditioned auto-decoder that represents a space of shapes. Its representation quality was improved by Duan et al. (2020) through curriculum learning.

Next, we discuss approaches closest to ours. MetaSDF (Sitzmann et al., 2020a) uses a meta network to predict the weights of an implicit SIREN network representing each shape. Wiesner et al. (2022) combine DeepSDF and SIREN to model the temporal evolution of growing and dividing *C. elegans* and lung cancer cells, which are morphologically much less complex than cortical neurons. De Luigi et al. (2023) train an individual, SIREN-based model for each object in the dataset and use its weights to predict a latent code, which forms the contextual input to an implicit decoder. This approach is impractical for large-scale datasets due to its high memory and compute costs.

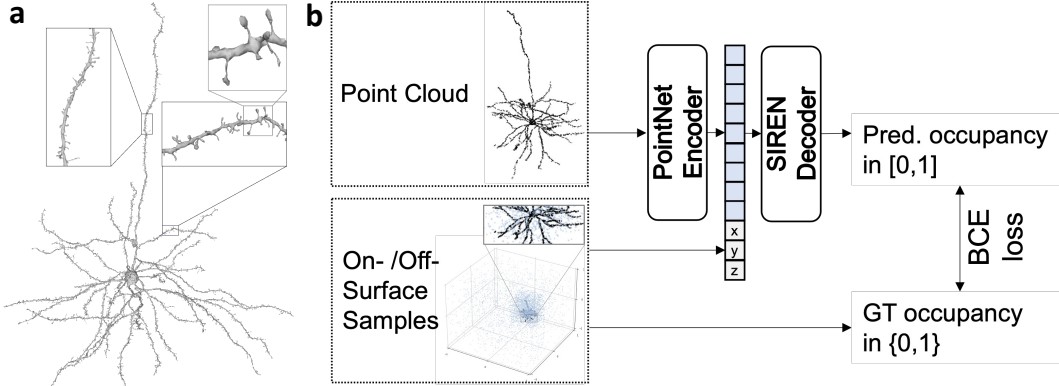

Figure 1: MORPHOCC. **a** 3D reconstructed mesh of a neuron in V1. Zoom ins show intricate details like spines on its dendrites. **b** Model architecture of MORPHOCC. The point cloud is encoded into a 64-dim latent vector. The decoder is an implicit model that predicts the occupancy of a sample in 3D space conditioned on the latent vector. The model is trained by optimizing the binary cross entropy between the predicted and the ground truth occupancy.

## 3 MORPHOCC

MORPHOCC is an implicit generative model to represent neuronal morphologies. The architecture consists of an encoder and a decoder (Figure 1). The encoder $g$ is a small PointNet (Qi et al., 2016), which encodes the input point cloud $\mathbf{P}$ into an embedding vector $\mathbf{z} = g(\mathbf{P})$. The input to the decoder is a 3D coordinate $x$ concatenated with the neurons embedding $\mathbf{z}$, and it outputs the probability of the point $x$ being inside the neuron's volume. Note, for a given neuron the decoder can be queried multiple times but the latent code $\mathbf{z}$ needs to be computed only once. We define the occupancy for a point $\mathbf{x} \in \mathbb{R}^3$ by the function $\varphi$ where $\varphi(\mathbf{x}) = 1$ if $\mathbf{x}$ is on the surface of the neuron (occupied) and $\varphi(\mathbf{x}) = 0$ if it is outside. We approximate the occupancy function $\varphi$ by a neural network $\hat{f}$:

$$\hat{\varphi}(\mathbf{x}) = f(\mathbf{x}, \mathbf{z}) = f(\mathbf{x}, g(\mathbf{P})). \tag{1}$$

Once trained, the surface of the neuron is implicitly represented by the zero iso-surface of $f(\mathbf{x}, \mathbf{z})$.

### 3.1 ARCHITECTURE

The encoder $g$ is PointNet (Qi et al., 2016) consisting of four encoder layers with 128, 256 and 512 units in the hidden layers and 64-dimensional output $z$. It uses batch normalization and ReLU activation functions.

The decoder $f$ follows SIREN Sitzmann et al. (2020b) and is an MLP with eight hidden layers, each with 512 hidden units. The network uses sine activation functions as nonlinearities, except in the last layer, where it uses a sigmoid function to predict the occupancy probability.

### 3.2 SAMPLING POINTS FOR THE IMPLICIT DECODER OF MORPHOCC

For training our model, we sample multiple 3D coordinates $x$ as input for each neuron in a batch. In each minibatch, we sample 5,000 points randomly from the surface of each neuron in this mini-batch. In addition, we sample 5,000 off-surface points. These are composed of 2,000 points drawn uniformly from within the volume containing all cells in the dataset. An additional 2,000 points are sampled uniformly within the tight bounding box of the neuron. The remaining 1,000 points are hard negatives, i.e. points that are close to the surface of the neuron, but outside of it. This way, our model learns the decision boundary between surface and non-surface of the neuron. To generate hard negatives, we sample a non-negative distance along the direction of the surface normals. The distance $d$ is defined as $d = \gamma\Delta + 10^{-3}$, where $\Delta$ is drawn from a log-normal distribution $\Delta \sim \mathrm{LogNormal}(0.002, 1)$ and $\gamma$ is a pre-factor that gets adjusted over training (next paragraph).

### 3.3 HARD-NEGATIVE-BASED CURRICULUM LEARNING

For training on hard negatives we use a curriculum strategy, where we progressively increase the level of difficulty. Specifically, we decrease the distance of the hard negatives to the neuron's surface by adjusting the factor $\gamma$ from initially 0.1 to 0.05 over the course of training. The parameters were chosen such that the distances $d$ approximately align with the typical thickness of a neuron's dendrites.

## 4 EXPERIMENTS

### 4.1 DATASET

We base our work on the MICrONS dataset (MICrONS Consortium et al., 2021), a $1.3 \times 0.87 \times 0.82 \,\text{mm}^3$ volume of tissue from the visual cortex of an adult P75–87 mouse. The volume has been densely reconstructed using serial section electron microscopy and has been further segmented into individual cells. It includes non-neuronal types and more than 54,000 neurons whose soma was located within the volume. It spans primary visual cortex (V1) and two higher visual areas, antero-lateral area (AL) and rostrolateral area (RL). We restrict ourselves to roughly $100 \,\mu\text{m}$ column across all cortical layers located in V1 that has been manually proofread and corrected for segmentation errors Schneider-Mizell et al. (2023). For this subset there are manual cell type labels available. We use these labels only to *evaluate* our model. We are not using them during training. We refer to the original papers on the dataset (MICrONS Consortium et al., 2021; Schneider-Mizell et al., 2023) and Appendix A.2 for further details on the identification and morphological reconstruction of individual neurons.

To generate the input to the encoder, we use the Trimesh library (Dawson-Haggerty et al., 2019) to sample points on the surface of the neuron. We sample twice as many points as there are vertices in the neuron's mesh (range ca. 152k–2.24M). For each point, we calculate the surface normal vector. We model only the dendritic morphology and remove the axons, because they are not reconstructed accurately for all neurons in this dataset. Preprocessing of the neurons' point clouds includes centering each neuron on its soma position and scaling (isotropically) by a constant factor across neurons such that all neurons lie within the unit cube $[-1, 1]^3$. We split the dataset into training ($n = 767$), validation ($n = 15$) and test set ($n = 15$). We use the test set for the neuron retrieval task ( subsection 5.3).

### 4.2 TRAINING

We use the Adam optimizer (Kingma & Ba, 2014) and a learning rate of $10^{-5}$. We train for 5.000 epochs with a minibatch size of 24 neurons. In each iteration, we sample 5,000 random points on the surface of the neuron and 5,000 off-surface points (details on the sampling procedure below). The on-surface points form the input to the PointNet encoder; both on- and off-surface points are used to train the implicit decoder.

The weights of the encoder are initialized uniformly using Kaiming initialization. For the decoder, the weights $W$ of the first layer are initialized uniformly between $\pm 1$, all others uniformly between $\pm\sqrt{6/(\omega_0^2 n)}$, where $n$ is the number of inputs and $\omega_0 = 30$ (see Sitzmann et al., 2020b). The loss function for training is the binary cross entropy on the occupancy predictions of the decoder.

### 4.3 BASELINES

We compare our reconstruction results to other model architectures that have been used to learn to represent a shape space of objects, namely DeepSDF (Park et al., 2019), Occupancy network (OccNet) (Mescheder et al., 2019) and the model proposed by Wiesner et al. (2022). In addition, we compare different encoders: SIREN (Sitzmann et al., 2020b), DGCNN (Wang et al., 2019) and Point-MAE (Pang et al., 2022) in combination with our implicit decoder. We focus on a comparison of encoder and implicit decoder architectures. All baselines are trained with the same cross-entropy occupancy loss.

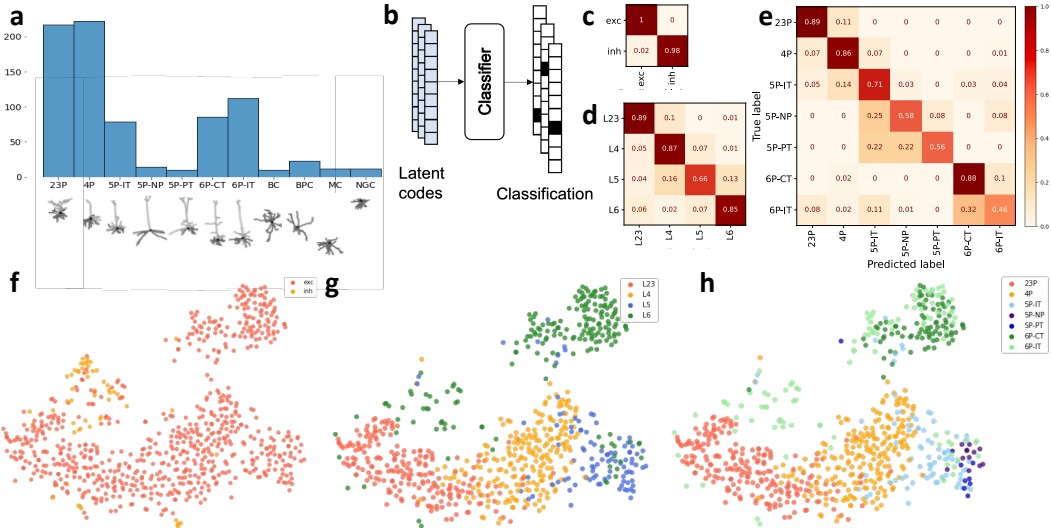

Figure 2: **Latent codes. a** Distribution of cell types in the subvolume of MICrONS Minnie. **b** Classifier trained on latent vectors of MORPHOCC. **c-e** Confusion matrices of classifier predictions on test sets of cross-validation for **c** coarse cell type, **d** layer and **e** cell type. **f-h** t-SNE embeddings (perplexity=30) of the latent codes colored by **f** coarse cell type, **g** layer of excitatory neurons and **h** cell type of excitatory neurons.

## 4.4 SURFACE RECONSTRUCTION AND VISUALIZATION

We reconstruct an explicit surface mesh from our implicit model using the Marching Cubes algorithm on the predicted occupancy of our decoder, decoding a 3D grid conditioned on the neuron's latent code of arbitrary resolution. To enhance the quality of our reconstructed meshes used for visualization, we remove small components using a greedy algorithm that progressively adds components until at least 75% of the vertices are included.

## 4.5 EVALUATION

We evaluate our model and the baselines using the established metrics and procedures by Mescheder et al. (2019). The following metrics are calculated based on the ground truth (GT) mesh and the predicted mesh. To evaluate the reconstruction results of our model, we calculate three metrics: volumetric Intersection over Union (IoU), mean Chamfer-$L_1$ distance (CD) and normal consistency (NC). Volumetric IoU is defined as the intersection of the predicted and GT volume, divided by their union. We sample 100k points, 50k points on the GT surface and 50k in the unit cube, and determine whether the points lie inside or outside the volume of each mesh. Chamfer distance is defined as the mean distance of the points in the predicted mesh to their nearest neighbors in GT. To estimate it, we sample 100k points from both meshes and estimate the distances between nearest neighbors using a KD-tree. Normal consistency measures how well the surface normals of both meshes align by calculating the mean absolute dot product of the normals and compare them to the normals of the corresponding nearest neighbors in the other mesh. The IoU metric is not very expressive in our context, because neurons occupy only an extreme small fraction of the volume compared to typical ShapeNet objects, for which the IoU metric was initially proposed (Mescheder et al., 2019). We therefore developed another metric, which we refer to as "local IoU", by sampling 100k points that are close to the neuron's volume. To this end, we first sample a point on the surface of the GT mesh and then add isotropic Gaussian noise. Half of these points are sampled in close vicinity of the neuron (SD 0.001), the other half further away (SD 0.01). The metric is then defined as the IoU of the predicted and GT occupancy of these points.

## 5 RESULTS

### 5.1 MORPHOCC'S LEARNED EMBEDDINGS CAPTURE MORPHOLOGICAL FEATURES WELL

The encoder embeds the full 3D shape of a neuron into a compact latent vector $\mathbf{z}$. We start by evaluating the nature of this learned latent space. The dataset contains 797 neurons, the majority of which are seven pyramidal neuron types, along with four types of interneurons (Figure 2**a**).

Qualitatively, the latent space captures the different cell types' morphological features when reduced to two dimensions using t-distributed stochastic neighbor embedding (t-SNE) (van der Maaten & Hinton, 2008) (Figure 2**f-h**). Inhibitory cells are predominantly grouped in the top-left region. There is a noticeable gradient from layer 2/3 to layer 4 pyramidal cells, which continues with layers 5 and then 6. Some of the layer 6 cells are more dispersed as they morphologically resemble inhibitory and more superficial cells, and the model is not provided with the laminar location. Layers 5 and 6 are further grouped into distinct cell types, which also cluster in latent space; except for 6P-IT cells, which are morphologically diverse and more dispersed.

To quantify the representative power of the latent space, we train a support vector machine (SVM) classifier on the latent codes to predict cell types and layer (Figure 2**b**). We followed the procedure described in (Weis et al., 2022) to find the best hyperparameters for our classifiers using ten-fold cross-validation on a grid search (details in A.3). Excitatory and inhibitory cells can be classified almost perfectly using a combination of our latent codes and the spine density as an additional input feature (Figure 2**c** and Table 1), substantially outperforming earlier work using the self-supervised features and spine density Weis et al. (2022). The latent codes also effectively capture the relative depth towards the pia, leading to accurate classification of excitatory neurons into specific layer boundaries (Figure 2**d**). The predictions for layer 2/3 and layer 4 exhibit high accuracy with only a few errors (Figure 2**e**). When it comes to layer 5 and 6 neurons, some degree of confusion arises. Layer 5 comprises three distinct cell types that share greater similarities among themselves than with neurons from other layers. Layer 6 IT cells are highly dissimilar, they differ in size and therefore some resemble more L6-CT neurons, the smaller ones more the layer 2/3 cells. Since the model has no information about the cortical depth, this result is reasonable.

We further demonstrate representational power of the latent codes on predicting the manual labels of coarse cell type (Inhibitory/Excitatory, I/E), cell type and layer and compare the results to GraphDINO, a recent representation model on neuronal morphologies Weis et al. (2022) (Table 1). We observe distinct strengths in their respective latent embeddings. Specifically, MORPHOCC's latent codes demonstrate

Table 1: Classification performance on latent codes. The values are the cross-validated balanced accuracy of out-of-bag classifiers.

|  | I/E | cell type | layer |
|---|---|---|---|
| GraphDINO | 92% | 85% | 89% |
| MORPHOCC | 99% | 74% | 84% |

superior performance in distinguishing between inhibitory and excitatory cells, displaying a substantial +7% gain in balanced accuracy in this regard, while being inferior to GraphDINO's latent codes on classification into cell type and layer. While the performance of our model is roughly in the same ballpark, this result suggests that some of the details about a neuron's shape that are contained in our model's embeddings are not directly helpful for cell type classification. This is perhaps not surprising, as reconstruction and classification are two very different objectives.

### 5.2 RECONSTRUCTIONS SHOW REPRESENTATIONAL POWER OF MORPHOCC

An important capability of our model is reconstruction, enabling the visualization of neurons' crucial structural and morphological features. We now turn to evaluating the reconstruction performance of our approach against a number of baselines. All models capture the rough outline and size of the neurons (Figure 3). However, the DeepSDF architecture fails to represent details in the individual dendrites of the neurons. The model of (Wiesner et al., 2022), which utilizes sine activation functions to represent high-frequency values, reconstructs significantly more details and fine-grained structures. OccNet, while capable of generating reasonable meshes, still falls short in terms of capturing intricate details. MORPHOCC stands out by capturing the most details in the shapes of individual dendrites within the neurons. While this achievement underscores our model's ability

Table 2: Quantitative 3D reconstruction measured using normal consistency (NC), Chamfer-$L_1$ distance (CD) in $\mu$m, volumetric intersection over union (IoU) and localized (volumetric) IoU.

|  | NC ↑ | CD ($\mu$m) ↓ | IoU ↑ | localized IoU ↑ |
|---|---|---|---|---|
| DeepSDF (Park et al., 2019) | 0.5630 | 21.18 | 0.9971 | 0.22 |
| Wiesner et al. (2022) | 0.5523 | 13.76 | 0.9987 | 0.23 |
| OccNet (Mescheder et al., 2019) | 0.5747 | 7.52 | 0.9948 | 0.25 |
| no encoder | 0.5999 | **2.37** | 0.9875 | 0.41 |
| Siren encoder (Sitzmann et al., 2020b) | 0.5970 | 4.91 | 0.9862 | **0.42** |
| DGCNN encoder (Wang et al., 2019) | 0.6014 | 3.48 | 0.9905 | 0.40 |
| Point-MAE encoder (Pang et al., 2022) | 0.5208 | 20.65 | 0.9820 | 0.28 |
| MORPHOCC | **0.6021** | 4.44 | **0.9997** | 0.33 |

to preserve fine structural details during the reconstruction process, there is still clearly room for improvement when comparing to the ground truth.

Our qualitative observations are supported by quantitative findings (Table 2): Different variants of our model achieve the best metrics. MORPHOCC with the simple Point-Net encoder achieves the highest normal consistency and Intersection over Union (IoU). In terms of Chamfer distance, more sophisticated encoders or directly learning the embeddings (auto decoding) achieve more fine-grained reconstruction than the simple Point-Net encoder. The IoU metric proposed by Mescheder et al. (2019) is not very informative in our setting, because neurons are extremely fine structures that occupy only a very small fraction of the volume, rendering almost all off-surface points far away from the surface and all models are above 0.98. We therefore computed an additional localized IoU, which focuses on points close to the neuron. This metric confirms that MORPHOCC outperforms the baselines and that the stronger DGCNN encoder or learned embeddings improve reconstruction quality. However, because the PointNet encoder resulted in qualitatively the best embeddings and is the simplest, we chose to use this version for further analysis despite its somewhat

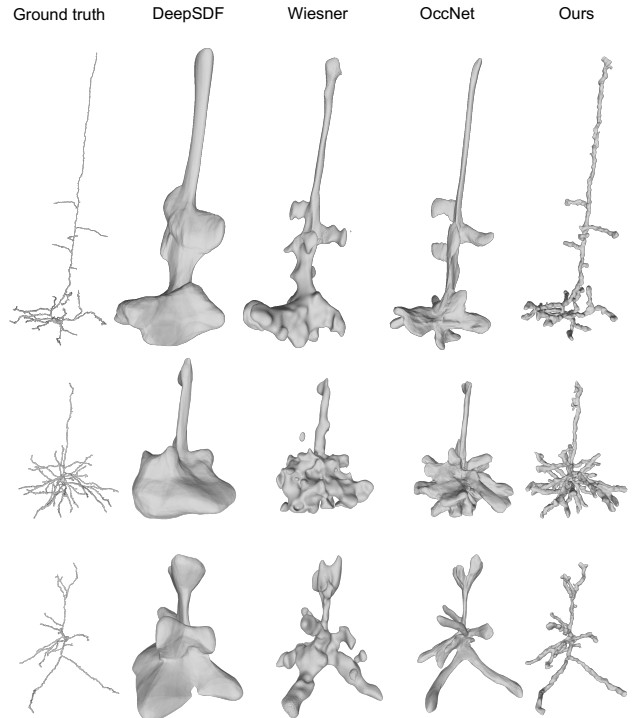

Figure 3: Qualitative 3D reconstruction results. First column: ground truth, following columns: reconstructions of various baselines, last column: reconstruction of our model.

weaker reconstruction quality. Directly learning the embeddings (no encoder) produces the most accurate reconstructions, but the embedding space was not organized semantically at all – the model essentially learned a lookup table and completely overfitted on the samples in the training set.

## 5.3 INFER LATENT CODES FOR UNKNOWN NEURONS

We use MORPHOCC to infer latent codes for unseen neurons. The encoder outputs a latent code which we use to classify the unseen neuron (test set). This functionality is valuable when new

neuron shapes become available, enabling us to classify them without the necessity of retraining the entire model. Moreover, this process serves as a testament to the model's generalization capabilities, as it effectively handles out-of-distribution samples.

Here we shown results for neuron retrieval. Given the latent code of an unseen neuron, we calculate its similarity to the latent codes of known neurons and retrieve the five nearest neighbors (NN) along with their respective labels. The label for the unseen neuron is assigned through majority voting among these retrieved neighbors. In Figure 4, we visualize three instances of neuronal retrievals, each presented in a row. Within this context, the blue box represents the unknown neuron, while the neurons in the respective row are the five nearest neighbors, accompanied by their labels below. The first row is labeled as layer 2/3 pyramidal neuron, because four out of five neighbors are identified as 23P cells. Only one cell is a layer 4 pyramidal neuron. Yet, it is crucial to note the discernible similarity between this L4 neuron and the unknown neuron. Remarkably, in subsequent examples, all retrieved neurons share the same label.

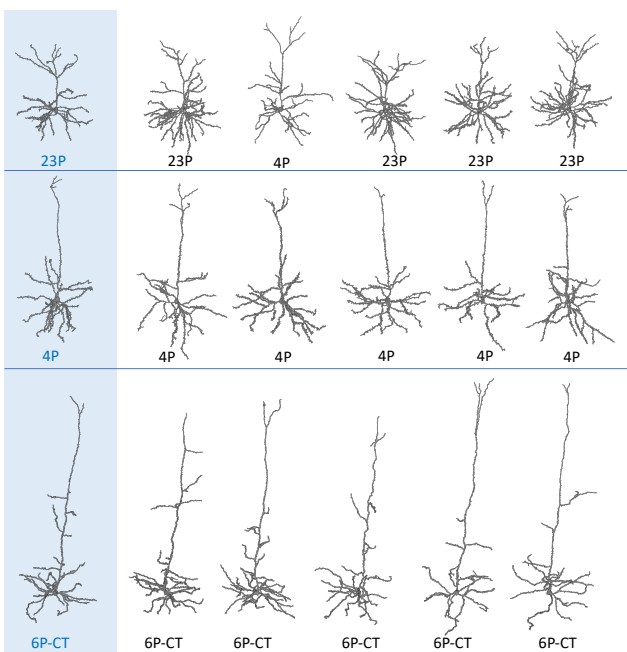

Figure 4: Neuron retrieval. Test set neuron (blue shaded) with inferred label along with five retrieved neurons based on the MORPHOCC embedding.

## 5.4 GENERATION OF NEURONAL MORPHOLOGIES

We use our model to generate new neurons based on latent codes. Generation essentially reflects our comprehension of the fundamental attributes that define a neuron. Our approach to generating neuronal morphologies involves interpolation between two distinct neurons. To delve deeper into this process, we interpolate between the latent codes $x_{n_0}$ and $x_{n_1}$ of two neurons $n_0$ and $n_1$, effectively generating intermediate latent codes that lie in between

$$x_\alpha = \alpha x_{n_0} + (1 - \alpha) x_{n_1}, \quad (2)$$

where $\alpha \in [0.2, 0.4, 0.6, 0.8]$. By querying the decoder with these interpolated latent codes $x_\alpha$ together with samples in a 3D grid, we obtain the predicted occupancy for these samples. We reconstruct the mesh as described in 4.5. This method enables us to not only generate new neurons but also to explore the continuum of neuronal morphologies lying between the two original examples.

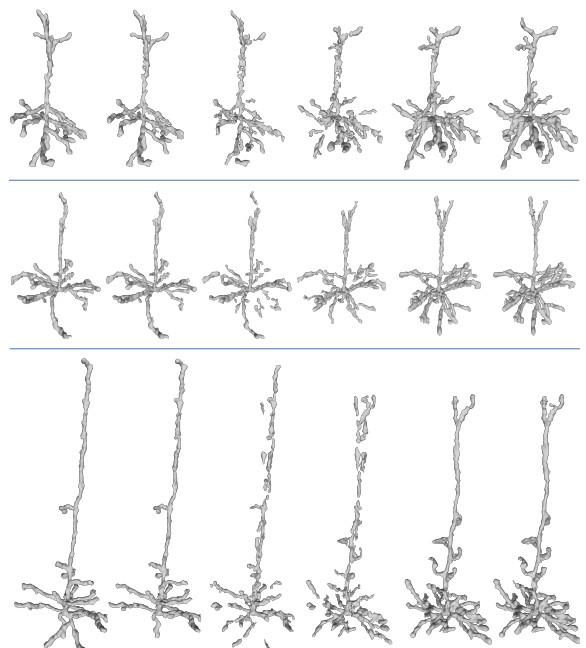

Figure 5: Interpolation series. First and last neuron are reconstructions, the four shapes in between are neurons generated by interpolating between those neurons.

Table 3: Ablation study on sampling strategy, curriculum learning and network architecture.

| Sampling strategy | NC ↑ | CD ($\mu$m) ↓ | IoU ↑ | local IoU ↑ |
|---|---|---|---|---|
| w/o perturbed | 0.5952 | 4.67 | **0.9998** | 0.30 |
| w/o uniform & perturbed | 0.5865 | 4.49 | 0.9951 | 0.31 |
| w/o restricting to bounding box & perturbed | 0.5426 | 12.04 | 0.9981 | 0.25 |
| only on surface (training diverged) | - | - | - | - |
| w/o curriculum learning | 0.5999 | 5.24 | 0.9991 | **0.34** |
| **Network architecture** | | | | |
| ReLU in decoder | 0.5267 | 34.93 | 0.9807 | 0.22 |
| shape dim = 32 | 0.5979 | 4.61 | 0.9997 | **0.34** |
| shape dim = 32 & hidden layers = 12 | 0.6008 | 4.47 | 0.9995 | 0.33 |
| **MORPHOCC** | **0.6021** | **4.44** | 0.9997 | 0.33 |

Figure 5 shows a series of interpolations. In each row, the first shape represents the reconstruction of a neuron from our training dataset, while the last shape represents the reconstruction of its neighboring neuron. In between the generated neurons that exemplify a continuum of morphological changes. In the first row, the shape of the neuron changes gradually from one to the other, characterized by the dissipation and regrowth of dendrites in alignment with the neighboring neuron. Notably, the basal dendrites become progressively denser throughout this transformation. The second row depicts a transition from an atufted neuron morphology to one with a small tuft. Finally, the third row showcases a similar transition, but in this case, the neuron undergoes significant shortening. Within this interpolation, the one oblique dendrite in the original neuron gradually dissipates, while multiple new obliques form below.

## 5.5 ABLATION STUDY

Finally, we test how different components of our model, point sampling strategy and training procedure influence the reconstruction performance of our model. All four aspects of our sampling strategy are necessary to achieve the best performance (Table 3), as does introducing the curriculum on the hard negatives during training. In terms of network architecture, we restricted the ablation on the top three architectures of an extensive hyperparameter search. The sine activation function in the decoder is very helpful, as expected from previous work Sitzmann et al. (2020b). Reducing the dimensionality of the embedding decreased performance only mildly, as did at the same time increasing the depth of the decoder (Table 3).

## 6 CONCLUSION

In this paper we introduced MORPHOCC, a model that learns vector representations of 3D neuronal morphologies while also being able to generate new morphologies. Our experiments demonstrate that the model enables the classification of neuronal morphologies into cell types based on the low-dimensional embeddings learned by the model. We show that the embeddings can be used to retrieve similar neurons and apply cell type labels to new neurons. Competitive methods learn the shapes of the neurons but fail to reconstruct the individual dendrites with their relative depth and curvature, while MORPHOCC succeeds in this task. However, our model is not yet able to reconstruct fine-grained details like spines and synapses. In the future, this limitation could be addressed by using a hierarchical, multi-scale model. In summary, our work provides a first step towards models that simultaneously embed and generate 3D neuron shapes which has the potential to improve our understanding of neurons in the brain.

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

# A  APPENDIX

## A.1  EXPERT CELL TYPE LABELS

A set of 1011 neurons was taken from a $100\mu m$ column in primary visual cortex and was labeled by visually inspecting individual cells (see Schneider-Mizell et al. (2023)). Cell types were assigned by considering morphology, synapses and connectivity, nucleus features and the cell's $(x, y, z)$ location. The cell types are layer 2/3 pyramidal (23P) and layer 4 pyramidal neurons (4P), layer 5 near-projecting (5P-NP), extra-telencephalic (5P-ET) (also know as pyramidal tract (5P-PT)) and inter-telencephalic (5P-IT) neurons, layer 6 intertelencephalic (6P-IT) and cortico-thalamic (6P-CT) neurons, Martinotti cells (MC), basket cells (BC), bipolar cells (BPC) and neurogliaform cells (NGC). The number of cells per cell type is visualized in Table 4.

Table 4: Cell type count for MICrONS dataset.

| layer | cell type | abbreviation | count |
|---|---|---|---|
| 2/3 | pyramidal | 23P | 217 |
| 4 | pyramidal | 4P | 222 |
| 5 | inter-telencephalic | 5P-IT | 79 |
| 5 | near-projecting | 5P-NP | 14 |
| 5 | pyramidal tract | 5P-PT | 10 |
| 6 | cortico-thalamic | 6P-CT | 86 |
| 6 | intertelencephalic | 6P-IT | 112 |
| - | basket cells | BC | 10 |
| - | bipolar cells | BPC | 23 |
| - | Martinotti cells | MC | 12 |
| - | neurogliaform cells | NGC | 12 |

## A.2  DATASET

The meshes were extracted from MICrONS explorer on version v374. We refer to MICrONS Consortium et al. (2021) for details on the MICrONS data. Proofreading includes several steps: The first step is decomposition into individual meshes per neuron. After this step there are so-called soma-mergers where two or more somata are merged falsely into one. The multi-soma are split in the second step. In the third step the axons and dendrites are identified. The last step is automatic proofreading. We refer to Celii et al. (2023) for more details on the proofreading procedure.

## A.3  SUPERVISED CLASSIFIERS

To find the best hyperparameters for the classifiers on coarse cell types, layers and cell types, we used ten-fold cross-validation on a grid search, following the procedure in (Weis et al., 2022). We report the accumulated test scores normalized by row. For classifying the neurons into coarse cell types, we trained a support vector machine (SVM). The best hyperparameters were: polynomial kernel of degree 2 and $C = 20$. The best classifier for layers and cell types is an SVM with polynomial kernel of degree 2 and $C = 1$ and using class weights.

