# OpenReview forum: "MorphOcc: An Implicit Generative Model of Neuronal Morphologies"
_ICLR.cc/2024/Conference — Submitted to ICLR 2024_

### Official Review · Reviewer_zrce · 2023-10-31

**Soundness:** 3 good
**Presentation:** 3 good
**Contribution:** 2 fair
**Rating:** 3
**Confidence:** 3

**Summary:**

- Morphocc is a generative model for soma + dendritic tree of cortical neurons.
 - The input consists of 3d point clouds derived volumetric reconstructions of neurons imaged with electron-microscopy.

 The model has two main components:
 1. A Pointnet learns a global representation vector per dendritic tree.
 2. An implicit decoder (SIREN) learns a function (conditioned on the global representation) that assigns whether a given position is on a given neuron or outside of it.

Authors use an EM microscopy dataset of ~800 neuron morphologies, and perform comparisons to justify architecture choice, share classification results to validate representations, and showcase reconstructions from latent traversals.

**Strengths:**

- S1. Manuscript is clearly written and easy to follow.
 - S2. The authors come up with a scheme of training that provides reasonable reconstructions of training set soma + dendite shapes from point clouds, with a relatively small number of neurons
 - S3. Authors perform detailed comparisons with various encoder and decoder choices

**Weaknesses:**

- W1. The method doesn't quite learn topology of neuronal dendrites. For example, in the latent traversals, the intermediate shapes are broken pieces and not valid dendritic trees (Fig 5). This is further exacerbated by post-processing.
> To enhance the quality of our reconstructed meshes used for visualization, we remove small components using a greedy algorithm that progressively adds com- ponents until at least 75% of the vertices are included.

 - W2. The volume bound has to be selected a-priori for the dataset. This approach seems to not be extensible for non-local morphologies (e.g. considering long range axons would require looking at the entire brain volume)

 - W3. It seems that the network has to be evaluated on points spread throughout the entire volume to create the iso-surface. This seems quite expensive to generate a single neuron's dendritic morphology.

 - W4. I think the following statement is a bit of an over-reach, since no tests are performed with out of distribution samples.
> Moreover, this process serves as a testament to the model’s generalization capabilities, as it effectively handles out-of-distribution samples.

 - W5. Overall, I think capturing connectivity (e.g. tree structure) should be a crucial ingredient in generative models for morphology. This is not captured in the current model. There isn't methodological novelty (Pointnets and SIREN are off-the-shelf components).

**Questions:**

- Q1. If one had 10x or even 100x as many morphologies, how would this method accommodate this? Is it mainly through increasing complexity of the decoder?

 - Q2. Could the authors elaborate on the no encoder experiment? Specifically, how is this aspect trained, and what embeddings is this referring to:
> Directly learning the embeddings (no encoder) produces the most accurate reconstructions, but the embedding space was not organized semantically at all

 - Q3. The soma position and coarse dendritic density are very suggestive of the cell types. In Fig 2, the representations seem organized by that. Is this a fair assessment? If so why is classification a good test of representations?

 - Q4. While not explicitly so, the volume is the same for the entire dataset, and the soma / origin of the dendrites contains layer information. Do the authors agree?
> Some of the layer 6 cells are more dispersed as they morphologically resemble inhibitory and more superficial cells, and the model is not provided with the laminar location.

 - Q5. A comparison of classification results with something simple e.g. density representation / PCA of the neuron point cloud (with similar bounding boxes and normalization as chosen for the model here) would help assess the improvements better.

---

> ### Author Response · Authors · 2023-11-14
>
> Thanks for the detailed feedback!
>
> W1: Your points on the limited generalization are spot on, and we agree. However, this reflects the state of the field: There are currently no quantitative, unbiased methods to capture the 3d morphology of neurons. There are currently also no established implicit methods that can handle generation of entire distributions of complex 3d shapes. Established methods from 3d computer vision either perform much worse than ours at generating fine-grained structures (Fig. 3) or do not address generalization at all, but overfit to individual scenes/objects. While there is obviously room for improvement, our approach clearly improves over the existing state of the art both qualitatively (Fig. 3) and quantitatively (Table 2). Hence, we consider it a valuable contribution.
>
> W2: That’s correct, but it seems to be necessary for any method. To model a highly non-local object, we need to consider a large volume which is going to be very sparse. Are we missing something? Can you give an example for a method where this wouldn’t hold?
>
> W3: That’s also correct, but there are standard and efficient ways of doing so (e.g. marching cubes). The big advantage is that we don’t have to store and evaluate this entire volume during training, enabling us to work with such a large volume in the first place.
>
> W4: That’s a wording error on our side. Thanks for catching it! They are not out-of-distribution samples, but held-out samples not seen during training. We will fix it.
>
> W5: You are right that PointNet encoder and SIREN are not novel. The novelty lies in the realization that the problem is not the architecture but the structure of the training data: For such fine-grained structures as neurons are, the established strategies for sampling off-surface points are insufficient, which is why we propose the hard negative sampling along with a learning curriculum, which we show to improve performance substantially.
>
> Q1: Yes, with more data one could increase the capacity in the encoder and decoder as well as increase the latent dimension to capture more details.
>
> Q3/Q4: Please note that for all neurons the soma is put at the origin (0,0,0) of the coordinate system. Hence, the absolute soma position is not available to the model. While for some neuron types the rough dendritic morphology is obviously helpful, that’s not the case for all of them (e.g. layer 6 neurons which are often short and similar to layer 2/3 or 4 neurons).
>
> Q5: Thanks for the suggestion! We will add a density-based baseline for comparison.
>
> Given the low score, you seem to be dissatisfied with either the approach in general or the performance in “absolute” terms. Could you elaborate on the main reasons for this low score and what we could do to address it?

---

> ### Comment · Reviewer_zrce · 2023-11-20
>
> I appreciate the detailed response - I share reasons for maintaining my initial score below.
>
> W1-a: I agree that there aren't universally accepted approaches for generating neuron morphologies; and a major issue that holds this back is lack of a metric to compare neuron shapes. Competing methods in Table 2. (e.g. DeepSDF, OccNet) are general proposals, not specifically intended for neuron morphologies. I appreciate the effort to define and quantify a _state of the art_ for neuron morphologies with such methods. However, if the sampled shapes don't look like neurons (e.g. in terms of topology) then the metric chosen to define state of the art itself becomes questionable, and progress along those lines seems not meaningful from a neuroscience perspective. In my opinion, developing the method to include topological constraints would help in this regard.
>
> W1-b and W5: If the authors are actually proposing Morphocc as a more general purpose method that captures detailed structures better, perhaps it may make sense to not be burdened by the complexity of neuron morphologies + low data availability, and instead pursue a different route to focus on and demonstrate the method's core capabilities.
>
> W2: I mainly had approaches like [Cuntz et al. 2010](https://journals.plos.org/ploscompbiol/article?id=10.1371/journal.pcbi.1000877), or even [Laturnus et al. 2021](https://proceedings.mlr.press/v139/laturnus21a.html) in mind, which don't require rely on densely querying a volume.
>
> W3: I'm concerned about how expensive the proposed method is. Would you agree that to include axons in this framework, (assuming axon diameter is about 1/10th that of a dendrite), the required sampling density would go up 100x? Are there any ways to mitigate this? A rough quantification of the computational cost would also help. I appreciate the clarification that the method doesn't need to evaluate the entire volume at the same time.
>
> Overall, I recognize this is a very hard problem, and appreciate the work that went into this. Nevertheless, I think the approach does not make it any clearer how to model neuron morphologies (W1-a and W1-b).
>
> There are outstanding issues with generalization (interpolations produce broken objects that aren't neurons) and overfitting. A secondary (and much less important issue) is the high cost of this approach (in accounting for larger volumes/ axons, in increasing capacity of implicit function networks with dataset size).
>
> The authors have shown that the proposed method captures fine-scale details of 3d shapes; whether this is a general improvement or a result of innovations to deal with low data availability in this particular dataset remains unclear to me.

---

> > ### Author Response · Authors · 2023-11-23
> >
> > Regarding Q5: As you suggested, we compared our classification results with a density representation of the neuron point cloud. We devised vertical histograms of the normalized and centered point clouds. By segmenting the y-axis into 64 bins, we generated an embedding of equivalent dimensionality as generated by our model. The results of density-based classification (99% I/E, 75% cell type, 88% layer) are comparable to those achieved by our model (99% I/E, 74% cell type, 84% layer).
> >
> > Note that this baseline is anything but trivial, as it uses the domain knowledge that the vertical density distribution contains a lot of information about the cell type. Our method has learned to extract this information without us explicitly providing it. Nevertheless, it suggests that our current (Pointnet-based) encoder might be too strong a bottleneck not capturing much of fine structure, likely leading to memorization in the decoder and impede generalization.
> >
> > We conclude that the contribution of our work is the strategy for achieving fine-grained reconstruction, but there remains room for improvement in the development of a more powerful encoder that captures more of the fine structure in the embeddings and improves generalization.

---

### Official Review · Reviewer_7kwZ · 2023-11-01

**Soundness:** 3 good
**Presentation:** 4 excellent
**Contribution:** 3 good
**Rating:** 6
**Confidence:** 4

**Summary:**

The paper proposes a generative model of mouse V1 neuron shapes built with a PointNet encoder and an optional SIREN (MLP) decoder predicting a volumetric occupancy map. The encoder embeddings are shown to capture semantically meaningful features of the neurons useful for cell type, polarity and layer origin classification, while the generated meshes compare favorably with baseline methods. The embeddings are also shown to be useful for neuron retrieval and kNN cell type classification.

**Strengths:**

- Baseline methods for neural morphology generation only operate on skeletons, whereas MorphOcc uses a richer volumetric representation.
- The paper is overall well written and very easy to read.
- Sec. 2 provides an excellent overview of related work.
- A clean set of proofread dendrites is used for experiments, sourced from one of the largest public volume EM datasets.
- Local IoU metric is proposed to compensate for limitations of IoU when applied to volumetrically sparse objects like neurons.
- Results convincingly show that the embedding captures semantically meaningful features of neurons.
- The proposed model is evaluated against different baseline models of shape generation, and the paper reports results for different encoder architectures.

**Weaknesses:**

- Only a single EM dataset is used in the paper. Have you considered other proofread volume EM datasets, particularly ones that are from different species, e.g. fruit fly?
- The interpolation results are not convincing in that the generated meshes seem to have disconnected components and thus not represent valid neurons.

**Questions:**

- How would the model scale for significantly larger neurons? Notably, axons can be much longer than dendrites, and fill space in an even sparser manner.
- Have you evaluated the impact of a larger latent dimension size?
- How were the ratios of points to sample from the different classes determined? (section 3.2)

---

> ### Author Response · Authors · 2023-11-13
>
> Thank you for your constructive feedback and acknowledging that our work is thoroughly evaluated!
>
> Dataset: These datasets are large, complex and so far not standardized, making it a substantial engineering effort to work with additional datasets from different labs/consortia. What added benefit would you expect from a second dataset? We believe the current dataset shows both the challenges of the type of data and the progress we have made.
>
> Interpolation: We agree that the interpolation results are not yet satisfying in absolute terms. However, this reflects the state of the field: There are currently no quantitative, unbiased methods to capture the 3d morphology of neurons. There are currently also no established implicit methods that can handle generation of entire distributions of complex 3d shapes. Established methods from 3d computer vision either perform much worse than ours at generating fine-grained structures (Fig. 3) or do not address generalization at all, but overfit to individual scenes/objects. While there is room for improvement, our approach clearly improves over the existing state of the art both qualitatively (Fig. 3) and quantitatively (Table 2). We are happy to state this limitation more clearly in the revision. Would you agree with this assessment? If not, where do you disagree or what would you suggest we do?
>
> Larger neurons: Good question, but hard to answer without being speculative. Currently we do not have such a dataset. But considering the current state of affairs, all methods (including ours) will likely struggle even more when considering larger volumes. This will become an important question once methods have evolved more and are more capable.
>
> Regarding your second and third question on the size of the latents and the ratio of sample points, we have started to run additional experiments and will update here as soon as we have the results.

---

> > ### Comment · Reviewer_7kwZ · 2023-11-22
> >
> > Thank you for the answer.
> >
> > I acknowledge the overheads inherent in analyzing additional datasets. I think the primary value of including samples from different species would be twofold: 1) it would show that the advantages of your method are not limited to mouse cortical neurons, and 2) it would be helpful for the neuroscience community to better understand what the performance of your approach might be across species. Note that the neuron shapes can be very different between e.g. drosophila and mouse. This would also partially address the larger/more complex neurons question. I could furthermore imagine that looking at the embeddings from different datasets might allow you to discover novel cross-species relationships, which would be a very interesting application of your work.
> >
> > I agree with your assessment of the state of the art in modeling complex 3d shapes, and I think clearly stating the limitations of the interpolated shapes in the text will be sufficient in this context.

---

### Official Review · Reviewer_ZykZ · 2023-11-01

**Soundness:** 2 fair
**Presentation:** 2 fair
**Contribution:** 2 fair
**Rating:** 3
**Confidence:** 4

**Summary:**

The authors of the paper introduce MORPHOCC, a neural network model designed to capture and represent the diversity of neuron morphologies in the mouse primary visual cortex (V1). The model encodes the morphology of each neuron into a low-dimensional embedding from which the 3D shape can be reconstructed. Trained on 797 dendritic shapes of V1 neurons, the model's embedding effectively captures morphological features, aiding in cell type classification. The model also enables the generation of new neuron instances through interpolation in the embedding space.

**Strengths:**

- The approach addresses the essential need for quantitative, unbiased methods to capture and represent the structural and morphological features of neurons.
- MORPHOCC's ability to reconstruct 3D shapes from a low-dimensional embedding offers potential benefits for representing and analyzing neuronal morphologies.

**Weaknesses:**

- The reliance on existing deep learning architectures like the PointNet encoder and SIREN decoder, without significant modifications or enhancements, raises concerns about technical novelty, especially considering the high standards expected for technical novelty in ICLR.
- The training dataset consists of only 797 neurons, raising concerns about the model's ability to generalize, especially when applied to classifying and generating new neurons outside this limited set. This is somewhat evident from the very high IoU scores and limited diversity of interpolated samples in Figure 5.
- Using linear interpolation in the embedding space to generate new neuron instances may not produce neurons distinct from those seen during training. Essentially, this method interpolates between two known neurons, resulting in a neuron that isn't morphologically much different from the original ones.

**Questions:**

- When generating neurons via interpolation, how do structural details, like dendrite branching and length, evolve?
- What specific measures were taken to prevent overfitting, especially given that directly learning the embeddings led to overfitting?
- How might these findings be used in real-world applications, like neuroscience research or medical diagnostics?

---

> ### Author Response · Authors · 2023-11-13
>
> Thanks for the feedback! Your points on the limited generalization are spot on, and we agree. However, this reflects the state of the field: There are currently no quantitative, unbiased methods to capture the 3d morphology of neurons. There are currently also no established implicit methods that can handle generation of entire distributions of complex 3d shapes. Established methods from 3d computer vision either perform much worse than ours at generating fine-grained structures (Fig. 3) or do not address generalization at all, but overfit to individual scenes/objects. While there is room for improvement, our approach clearly improves over the existing state of the art both qualitatively (Fig. 3) and quantitatively (Table 2). Hence, we consider it a valuable contribution. Do you agree with this assessment? If not, where do you disagree?
>
> Regarding your specific questions and comments:
>
> Technical novelty: You are right that PointNet encoder and SIREN are not novel. The novelty lies in the realization that the problem is not the architecture but the structure of the training data: For such fine-grained structures as neurons are, the established strategies for sampling off-surface points are insufficient, which is why we propose the hard negative sampling along with a learning curriculum, which we show to improve performance substantially.
>
> Evolution of structural details when interpolating: That’s a great question! We will prepare a more in-depth analysis of this question for the final version.
>
> Overfitting: To prevent overfitting, we introduced an encoder, which acts as a bottleneck to organize the latent embeddings. In addition, we limited the complexity of the decoder and the size of the embeddings.
>
> Real-world applications: The learned vector embeddings are extremely useful as feature representations of neurons for revealing new cell types (e.g. [1]), for comparing cells across brain areas (e.g. [2]), species or diseases and for relating neuronal structure and function. In the context of scientific investigation, the generation capability is crucial for interpretability –  to understand what the abstract learned features mean and also what aspects/level of detail of neuronal morphology they capture.
>
> [1] Weis et al, bioRxiv 2022.  https://www.biorxiv.org/content/early/2022/12/22/2022.12.22.521541
>
> [2] Scala et al., Nature 2019. https://www.nature.com/articles/s41467-019-12058-z

---

> > ### Comment · Reviewer_ZykZ · 2023-11-20
> >
> > I would like to thank the authors for clarifying my earlier queries. I want to acknowledge the complexity and significance of the problem that the paper addresses. While I recognize the merit in the approach of using a bottleneck layer for grouping encodings and a simplified decoder to mitigate overfitting, I am concerned about the adequacy of this method given the complexity of the 3D problem at hand. In my view, robustly tackling such a complex issue requires a much larger and diverse dataset. Furthermore, my concerns regarding the technical novelty and the potential for overfitting in the study persist. These aspects are crucial in assessing the overall contribution and impact of the research. Given these considerations, I find it appropriate to maintain my initial rating of the paper.

---

### Meta-Review · Area_Chair_6tHg · 2023-12-13

**Metareview:**

The authors develop a novel implicit neural representation of neuronal morphology. This is a promising approach to scale up the representation of neuronal morphologies beyond classic dense voxel or sparse mesh representations, with the possibility of representing entire neuronal morphologies in a scalable manner. Reviewers expressed concerns with the limited evaluation of the method on the small MiCRONS dataset. Some also expressed concerns of limited novelty, given the use of existing pointnet and SIREN network architectures. Regardless of neural network architecture, this work represents a qualitative advance and a good first step towards implicit neural representations of neuronal morphology. However, the weakness of the empirical evaluation of the method could be improved.

**Justification For Why Not Higher Score:**

Weak empirical evaluation on a small single MiCRONS dataset. Limited generalization, perhaps due to small training dataset?

**Justification For Why Not Lower Score:**

NA

---

### Decision · Program_Chairs · 2024-01-16

Reject